# Evaluation of a Novel Prototype for Pressurized Intraperitoneal Aerosol Chemotherapy

**DOI:** 10.3390/cancers12030633

**Published:** 2020-03-09

**Authors:** Hee Su Lee, Junsik Kim, Eun Ji Lee, Soo Jin Park, Jaehee Mun, Haerin Paik, Soo Hyun Oh, Sunwoo Park, Soomin Ryu, Whasun Lim, Gwonhwa Song, Hee Seung Kim, Jung Chan Lee

**Affiliations:** 1Interdisciplinary Program in Bioengineering, Seoul National University Graduate School, Seoul 08826, Republic of Korea; heesulee@snu.ac.kr (H.S.L.); kimjunsik@snu.ac.kr (J.K.); 2Department of Obstetrics and Gynecology, Seoul National University Hospital, Seoul 03080, Republic of Korea; bliss880103@gmail.com (E.J.L.); soojin.mdpark@gmail.com (S.J.P.); jhee1315@gmail.com (J.M.); haerin.paik@gmail.com (H.P.); soohyun27oh@gmail.com (S.H.O.); 3Institute of Animal Molecular Biotechnology and Department of Biotechnology, College of Life Sciences and Biotechnology, Korea University, Seoul 02841, Republic of Korea; sunwoojump@korea.ac.kr (S.P.); tnals6917@korea.ac.kr (S.R.); ghsong@korea.ac.kr (G.S.); 4Department of Food and Nutrition, Kookmin University, Seoul 02707, Republic of Korea; wlim@kookmin.ac.kr; 5Department of Biomedical Engineering, College of Medicine and Institute of Medical and Biological Engineering, Medical Research Center, Seoul National University, Seoul 03080, Republic of Korea

**Keywords:** pressurized intraperitoneal aerosol chemotherapy, peritoneal carcinomatosis, nozzle, drug distribution, aerosolization, penetration depth

## Abstract

Pressurized intraperitoneal aerosol chemotherapy (PIPAC) has been suggested as an alternative option for treating peritoneal carcinomatosis (PC). Even with its clinical advantages, the current PIPAC system still suffers from limitations regarding drug distribution area and penetration depth. Thus, we evaluated the new PIPAC system using a novel prototype, and compared its performance to the results from previous studies related with the current MIP^®^ indirectly because the system is currently not available for purchase in the market. The developed prototype includes a syringe pump, a nozzle, and controllers. Drug distribution was conducted using a methylene blue solution for performance test. For penetration depth evaluation, an ex-vivo experiment was performed with porcine tissues in a 3.5 L plastic box. Doxorubicin was sprayed using the novel prototype, and its penetration depth was investigated by confocal laser scanning microscopy. The experiment was repeated with varying nozzle levels from the bottom. The novel prototype sprays approximately 30 μm drug droplets at a flow rate of 30 mL/min with 7 bars of pressure. The average diameter of sprayed region with concentrated dye was 18.5 ± 1.2 cm, which was comparable to that of the current MIP^®^ (about 10 cm). The depth of concentrated diffusion (DCD) did not differ among varying nozzle levels, whereas the depth of maximal diffusion (DMD) decreased with increasing distance between the prototype and the bottom (mean values, 515.3 μm at 2 cm; 437.6 μm at 4 cm; 363.2 μm at 8 cm), which was comparable to those of the current MIP^®^ (about 350–500 μm). We developed a novel prototype that generate small droplets for drug aerosolization and that have a comparably wide sprayed area and depth of penetration to the current MIP^®^ at a lower pressure.

## 1. Introduction

Solid tumors with peritoneal carcinomatosis (PC) show poor prognosis due to frequent relapses. PC is observed in 15%–80% of patients with ovarian and colorectal cancers at the diagnosis of recurrence, and the life expectancy is less than 20 months even after surgery and chemotherapy [1,2]. For better prognosis of PC patients, intraperitoneal chemotherapy (IP) and hyperthermic intraperitoneal chemotherapy (HIPEC) have been introduced [3,4,5]. IP works by directly administering chemotherapeutic drugs into the abdominal cavity and allowing them to penetrated tumor cells by diffusion. HIPEC may accelerate cell death of fragile tumor cells subjected to hyperthermia. However, both IP and HIPEC have limitations concerning drug distribution and penetration depth. The two methods have demonstrated uneven drug distribution across the abdominal cavity and small penetration depth, limiting the treatment of residual tumors after cytoreduction [6,7,8]. Moreover, HIPEC has a risk of renal or hepatic complications from high drug concentrations and hyperthermia [9].

On the other hand, pressurized intraperitoneal aerosol chemotherapy (PIPAC) has been introduced and has overcome these limitations of IP and HIPEC [10]. In PIPAC, chemotherapeutic drugs are aerosolized into the abdominal cavity via micropump nozzle (MIP^®^; Capnomed GmbH, Villigendorf, Germany) using a high-pressure injector [11]. As the aerosolized particles float around the cavity, the drug distribution area increases accordingly. Additionally, the drug particles penetrate tumor tissues more effectively as a result of gas influx at the pressure of 12 mmHg [12], which allows for the use of low-dose chemotherapeutic drugs unlike IP and HIPEC [5,13]. As described above, PIPAC has shown improvement in anti-cancer effects and reduced side effects during IP. Therefore, PIPAC has been suggested for effective treatment of PC with relatively low renal and hepatic toxicities using low-dose chemotherapeutic drugs and normothermia [12,13]. 

Although PIPAC is performed in nearly 30 countries, its availability has been limited despite its gradual increase in clinical demand. Therefore, we have developed a novel prototype that can successfully generate and spray aerosolized drug particles in the abdominal cavity. While a direct comparison of the prototype to the current PIPAC system would be ideal, the system is currently not available for purchase in the market. Thus, we have performed standard procedures to evaluate the performance of the nozzle, and a relevant ex-vivo experiment was conducted under the same conditions as previous PIPAC studies. Thereafter, we evaluated the effectiveness of our prototype by comparing its performance to the results from previous studies related with the current MIP^®^ indirectly.

## 2. Results

### 2.1. Granulometric Analysis and Spray Angle

The granulometric analysis was performed. The average number of aerosol particles detected and analyzed was 8168. Two types of mean diameter measurements, arithmetic mean diameter and Sauter mean diameter, were measured to test the nozzle performance. The Sauter mean diameter measurement is widely used to characterize a nozzle. The arithmetic mean diameter (D_10_) obtained from Equation (1) is the average of the diameters of all the particles, and Sauter mean diameter (D_32_) from Equation (2) is obtained from the volume to surface ratio. The Arithmetic and Sauter mean diameters of our nozzle were 25.4μm and 32.1μm, respectively as shown in Appendix A. The mean particle velocity from the nozzle was calculated as 1.31 m/s. The calculated spray angle is 77.2° with a flow rate of 30 mL/min (Figure 1A).
(1)$$D_{10}=\frac{1}{N}\overset{N}{\underset{i=1}{\sum }}n_{i}D_{i}\;$$
(2)$$D_{32}=\overset{N}{\underset{i=1}{\sum }}n_{i}D_{i}{}^{3}/\overset{N}{\underset{i=1}{\sum }}n_{i}D_{i}{}^{2}$$

### 2.2. Distribution Analysis with Methylene Blue Solution 

The sprayed area consists of a concentrated zone and a spread zone as shown in Figure 1B. We defined the concentrated zone as the area where the methylene blue solution is sprayed intensively. The spread zone describes the region where the solution is sprayed with less intensity. The terms “concentrated” and “spread” zones were not quantitatively described in this experiment. We have used the terms to avoid overestimation of the diameter of the sprayed area. Therefore, the actual distribution area of drug particles may be wider. When the nozzle orifice was kept 12 cm apart from a carton paper, the average diameter of the concentrated zone was 18.5 ± 1.2cm, and the average diameter of the spread zone was 28.3 ± 1.6cm. In the previous study where MIP^®^ was positioned at a distance of 15 cm, the intense dye inner area, similar to the concentrated zone in the current study, was about 10 cm [11].

### 2.3. Ex-Vivo Experimental Analysis 

Figure 2A shows depth of maximal diffusion (DMD) and depth of concentrated diffusion (DCD) after spraying the solution of doxorubicin by using confocal laser scanning microscopy. The mean values of DCD at nozzle positions of 2, 4 and 8 cm were 255.3 ± 4.5, 251.7 ± 9.5, and 253.1 ± 5.3 μm, respectively, without significant differences among groups (*p* > 0.05) (Figure 2B). On the other hand, the mean values of DMD were 515.3 ± 5.7 μm at 2 cm, 437.6 ± 3.6 μm at 4 cm, and 363.2 ± 7.4 μm at 8 cm. There was a significant decrease in DMD with increasing distance between the nozzle and the tissue (2 vs. 4 cm, 2 vs. 8 cm, and 4 vs. 8 cm; *p* < 0.05; Figure 2C). In the previous study where doxorubicin penetration was evaluated at different positioning of MIP^®^ to the bottom, the penetration depth, similar to DMD in the current study, range from 350 to 500 μm [14].

## 3. Discussion

PIPAC shows its effect on disseminated tumors of the parietal and visceral peritoneum through direct diffusion of anti-cancer drugs. It has been applied as palliative treatment to increase the quality of life with low toxicity in refractory solid tumors. In refractory ovarian cancer, it has shown clinical response of 62–88% and medial survival time of 11–14.1 months. Moreover, tumor response was 50–91% and 71–86%, and medial survival time was 8.4–15.4 months and 15.7 months, in patients with refractory gastric and colorectal cancers, respectively [15]. The most important factor of these effects is the proper distribution of anti-cancer drugs on disseminated tumors resulted from aerosolization, which suggests that the nozzle in PIPAC system is the key point for this treatment. 

The developed prototype in this study sprays approximately 30 μm drug droplets at a flow rate of 30 mL/min. The current PIPAC system produces the average particle size of 20 μm with a pressure of up to 20 bars [11,16]. While our prototype produces a slightly larger average particle size, 32 μm, our system is using pressure only up to 7 bars. This demonstrates that our prototype needs significantly lower pressure to generate the targeted particle size. We decided to limit our particle size to be slightly larger than aerosol particles in order to prevent possible drug leakage from aerosolization and thereby maintain a safe environment for clinicians and patients.

The drug distribution area was investigated when evaluating prototype performance. Using the same methylene blue solution used in a previous study [11], we compared the distribution area of our prototype to that of the current PIPAC system. As a result, our prototype demonstrated a comparable wide dyed area even at a shorter distance of 12 cm, suggesting that it would result in a wider drug distribution area in the abdominal cavity.

Moreover, we evaluated the penetration depth of drug in the tissue placed directly below the nozzle at varying nozzle levels in order to investigate the effect of nozzle height on penetration depth as the next step. As a result, the mean values of DMD were significantly different between groups. Also, this aligns with the reported penetration depths from MIP^®^ at the same nozzle positions of 2, 4 and 8 cm [11]. This indicates that our prototype has comparable performance as the current system regarding penetration depth. Moreover, it is meaningful that we have achieved these results only with 7 bars of pressure while the current system requires 20 bars of applied pressure.

While previous studies have only considered maximum penetration depth, we also investigated DCD values. It is significant to consider DCD values since the area with concentrated population of cells affected by drugs is more likely to promote cancer treatment rather than the area with few affected cells. The maximum penetration depth demonstrates only a few cells affected by the drug, and therefore the concentrated penetration depth was included in our study. Our results demonstrate comparable DCD at all nozzle levels with no significant differences. The similar DCD values at different nozzle positions (2, 4, and 8 cm) indicate that the nozzle level is not an important factor that determines the drug penetration depth in PIPAC. It is more crucial to expand the area of tissue directly exposed to the spray jet of the nozzle in order to promote homogeneous drug distribution for successful cancer treatment.

However, limitations still exist in the clinical setting. In particular, the drug distribution area by PIPAC will be expected to be not homogenous in the abdominal cavity. Previous studies have investigated the efficacy of an MIP^®^ nozzle for the PIPAC system through ex-vivo experiments, where the drug penetration depth was effective only in the tissue directly below the nozzle. They demonstrated poor penetration depth in tissues on the sidewalls and above the nozzle tip [14,17]. These findings demonstrate that the current PIPAC system may not produce aerosolized drugs freely moving in the abdominal cavity, and drug distribution should be improved for better clinical outcomes. 

PIPAC technology has focused on drug aerosolization for reduced particle size in order to promote homogeneous distribution. Angio-injectors used for high-pressure generation in the current PIPAC system deliver a pressure of up to 20 bars to a nozzle for 20 μm drug particles. While researchers are aiming for particle reduction by improving the nozzle technology, the technical difficulty is paramount as it requires higher pressure than the currently used amount. Additionally, such large pressure generation would result in an unstable system concerning its safety aspect and would require a huge amount of time, space and expenses. Therefore, the research focus of the PIPAC technology should be shifted towards expanding the drug distribution area directly exposed to the nozzle’s jet stream for better treatment outcomes. 

Mechanical movement of a nozzle can be suggested as a possible solution for increase in drug distribution area. The technology could overcome the current weaknesses of the PIPAC system by improving the distribution area while maintaining the drug penetration depth. Additionally, safety issues can be eliminated as it does not require high-pressure generation and therefore lower the risk of drug leakage due to small particle size. 

## 4. Materials and Methods

### 4.1. Prototype Development

The syringe pump was designed using a stepper motor (A200K-G599W-G10, Autonics, Korea), a syringe holder, and a ball screw. The stepper motor used for the pump generates a maximum torque of 19.6 N⋅m and generates 7 bars of pressure required for drug delivery. A ball screw driven linear actuator was connected to the stepper motor, and they were then secured to three blocks of the syringe pump. As shown in Figure 3A, the three blocks consist of an endplate, a pusher block, and a holder plate. The endplate holds the stepper motor, and the pusher block and holder plate are designed as a bracket for a 200 mL syringe. All blocks were CNC milled with aluminum and brass for high-pressure endurance. Four rods were used to connect the blocks and served as a guide rail. Figure 3B shows the assembly of the pump. A syringe can be inserted and snapped into the retainer brackets incorporated in the pusher block and holder plate. When the actuator is driven, the pusher block travels along the rods and pushes the syringe end for drug delivery. We incorporated a cap over the syringe to prevent bending and breakage of the syringe from the pressure. Figure 3C shows the 3D drawing of the nozzle. The nozzle was designed to go through a 12 mm trocar and therefore has a diameter of 10 mm. Fluid enters the nozzle through the two holes in the sleeve (1) which was designed to reduce the liquid volume remaining in the body after injection. The liquid then travels through the empty compartment (2) that meets with a pushing spring (3). As the liquid is pushed through the spring, fine particles are generated and travel through the groove (4). The particles enter the chamber between (4) and the nozzle tip (5), and turbulence is formed. As the particles leave the orifice and come into contact with the air, they become finer particles. Also, they are sprayed with a wide-angle as a result of the centrifugal force and high rotational speed of particles caused by the turbulence in the chamber. Two controllers were developed to drive the stepper motor (Figure 3D). The main controller consists of two Arduino UNOs, one controlling the syringe pump and the other receiving values from a load cell, an HC-06 Bluetooth module, and a touch screen (NX8048T070_011R, Nextion, USA). The remote controller is made of an Arduino UNO, a touch screen (NX8048T070_011R, Nextion, USA), an HC-06 Bluetooth module and a 9V battery. 

Shown in Figure 4, the signal from the remote controller is transmitted to the main controller via Bluetooth. It is then delivered to the stepper motor through a motor driver (MD5-HD14, Autonics, Korea). A load cell (TAS606, HT Sensor Technology Co., LTD, China) was attached to the pusher block to measure the force when the syringe moves forward to deliver fluids to the nozzle. Pressure applied to the nozzle is calculated from the measured value of force and the area of the syringe. The touch screen displays the pressure value along with the drug volume, flow rate and remaining procedure time. 

### 4.2. Granulometric Analysis and Spray Angle

Particle Dynamics Analysis (FiberPDA receiver, Dantec Dynamics, Denmark) was performed at the Korea Institute of Machinery and Materials to measure particle velocity and its mean diameter. 20 °C deionized water was filled in a syringe and sprayed via the nozzle. The particle size and velocity were measured 12 cm away from the orifice (Appendix A). A laser was used to measure the angle of the spraying nozzle. It was projected 12 cm apart from the orifice. The path of the laser was only shown where the particles were present. A picture of the side view of the spraying particles and the laser path was taken. The angle was derived from the distance between the orifice and the visible length of the laser path using the Pythagorean theorem.

### 4.3. Distribution Analysis with Methylene Blue Solution

The distribution property of the nozzle was performed by measuring the area with a methylene blue solution. The nozzle was set perpendicular to a carton paper by a clamp. A 30 mL of methylene blue solution was sprayed at 30 mL/min. The nozzle orifice was kept 12 cm apart from a carton paper. 

We measured the diameter of the sprayed area marked by methylene blue. The procedure was repeated five times to obtain a mean value. 

### 4.4. Ex-Vivo Experiment

An ex-vivo experiment was set up as shown in Figure 5. A 3.5 L hermetic plastic box was used to mimic the abdomen. Two 12 mm and 5 mm trocars (TR12F, TR05F, DalimSurgNET Co., Ltd, Korea) were inserted in the cover of the plastic box, and the gaps were completely sealed. The nozzle and temperature/humidity sensor (ETH-01D, Econarae, Korea) were inserted in 12mm, 5mm trocars respectively. Temperature (°C) and humidity (%) were displayed via a 16 × 2 character dot-matrix LCD module. CO_2_ supply was connected to the 5 mm trocar to maintain the pressure of 12 mmHg and the temperature of the plastic box was kept at 36 °C to simulate the environment of the abdomen during the PIPAC procedure. Three different plastic boxes and 2 × 2 cm peritoneal tissue samples were prepared prior to the experiment to study the effect of varying nozzle positions on the penetration depth. The investigated distances between the nozzle and the tissue were 2, 4, and 8 cm. For each group, a tissue sample was placed on the opposite side of the nozzle tip and sprayed with doxorubicin. The solution was prepared by diluting 3 mg doxorubicin in 50 mL 0.9% NaCl at room temperature (23 °C). After spraying the solution, the tissue sample remained in the box for 30 minutes while maintaining the same pressure and temperature. After 30 minutes, the tissues were rinsed with a 0.9% NaCl solution to remove unbound, superficial cytostatic substances and were frozen in liquid nitrogen. We obtained 10 µm cryosection and mounted them with 1.5 µg/mL 4′,6-diamidino-2-phenylindole (DAPI) (Sigma-Aldrich) to stain nuclei. The penetration depth of doxorubicin was evaluated with three samples from each tissue using confocal laser scanning microscopy (Leica TCS SP8). The DMD was calculated as the distance between the sample’s luminal surface and the innermost depth at which positive staining for doxorubicin could be visualized. The DCD was calculated as the distance between the sample’s luminal surface and the surface where the positive staining is most accumulated.

### 4.5. Statistical Analysis

We compared DMD and DCD among different nozzle heights using the Kruskal-Wallis test with a post-hoc Mann-Whitney U test including the Bonferroni correction. We used SPSS software version 21.0 (SPSS Inc., Chicago, IL, USA), and a p-value less than or equal to 0.05 was considered statistically significant.

## 5. Conclusions

We developed a novel prototype that generates small droplets from drug aerosolization and a comparably wide sprayed area and depth of penetration compared to the current MIP^®^ at lower pressure. Therefore, this novel prototype is expected to show comparable effects for PIPAC, which should be investigated in further studies. 

## 6. Patents

H.S.L., J.K., S.J.P., H.S.K. and J.C.L. have a Korean pending patent entitled “Apparatus for spraying drug” (KR1020190125181).

## Figures and Tables

**Figure 1 cancers-12-00633-f001:**
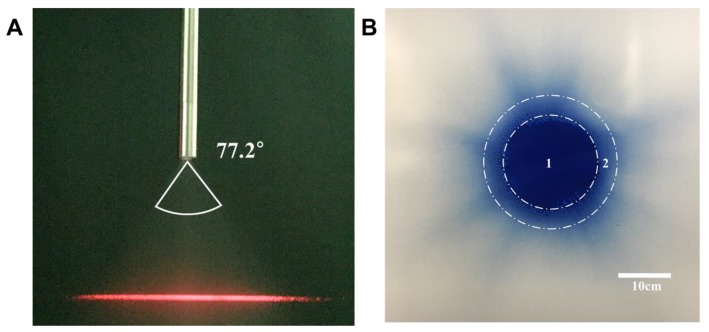
(**A**) Spray angle derivation. The spray angle of the nozzle is 77.2°. (**B**) Analysis of drug distribution with methylene blue solution. Circle 1 and 2 show the concentrated and spread distribution areas, respectively. The average diameter of the concentrated zone was 18.5 ± 1.2 cm.

**Figure 2 cancers-12-00633-f002:**
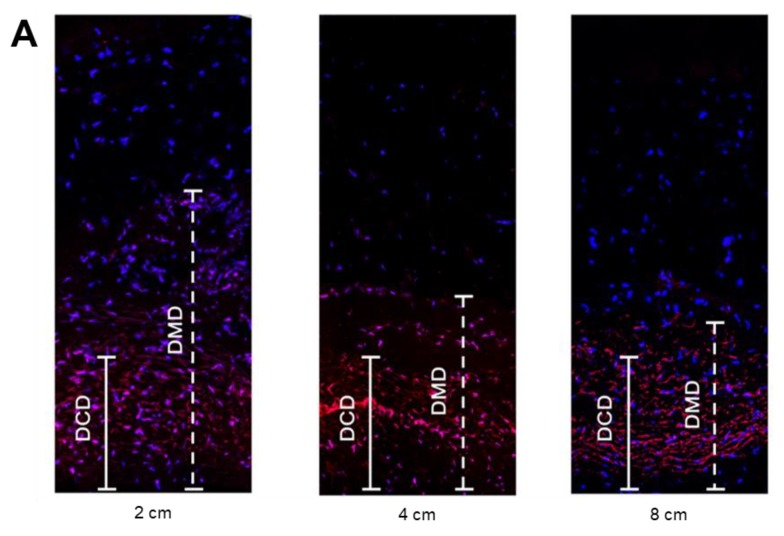
Confocal laser scanning microscopy analysis; (**A**) depth of concentrated diffusion (DCD) and depth of maximal diffusion (DMD) in tissues treated with doxorubicin at different nozzle levels; (**B**) no difference of DCD among nozzle levels of 2, 4 and 8 cm; (**C**) decreased of DMD with increasing nozzle levels (* *p* < 0.05).

**Figure 3 cancers-12-00633-f003:**
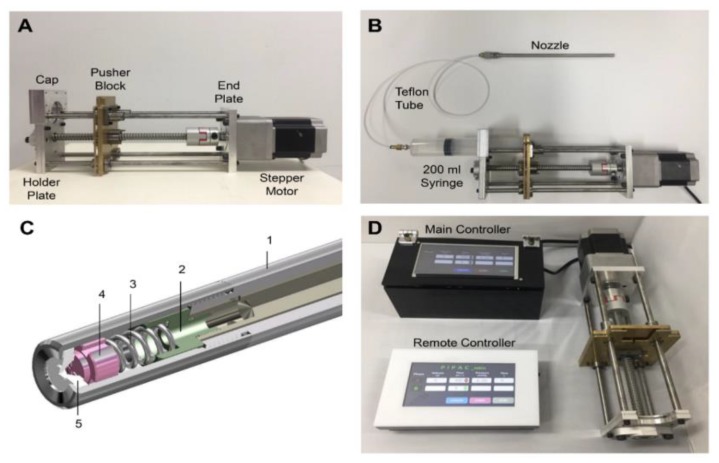
(**A**) A novel prototype of the new pressurized intraperitoneal aerosol chemotherapy (PIPAC) system using a stepper motor; (**B**) the new PIPAC system assembled with a 200 mL syringe and a nozzle; (**C**) 3D cross sectional view of the developed nozzle head; (**D**) the main and remote controllers used for device operation.

**Figure 4 cancers-12-00633-f004:**
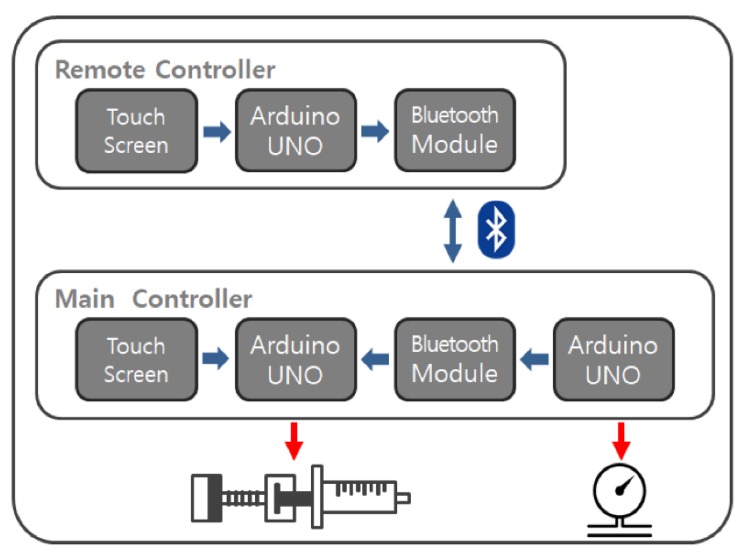
Schematic diagram of the prototype. The controllers communicate with each other via Bluetooth. The main controller operates the syringe pump and receives force values from the load cell.

**Figure 5 cancers-12-00633-f005:**
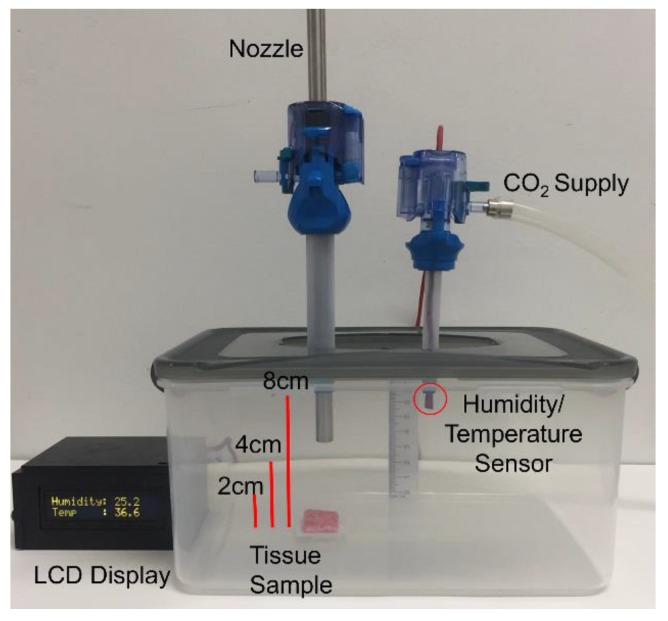
An ex-vivo experiment was conducted at varying nozzle levels representing the distance between the nozzle and the tissues at bottom (2, 4, and 8 cm).

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
