# Peer review of "Evaluation of a Novel Prototype for Pressurized Intraperitoneal Aerosol Chemotherapy"

_cancers, 2020, doi:10.3390/cancers12030633_

Round 1

Reviewer 1 Report

Review for Manuscript cancers-719764-peer-review-v1

General Comments: Overall, all sections are very well-written and designed to describe this novel apparatus. No major comments. I have a few specific comments that are listed below by section and line number.

More Specific Comments:

Abstract

Line 28 – Remove “for performance test”

Introduction

Line 49 – Change “penetrated” to “penetrate”

Results

Line 79 – Reword/rephrase “average of 8168 particles”. I am having difficulty understanding this sentence

Discussion – None

Materials and Methods – None

Conclusions – None

Figures and Tables – None

Author Response

REVIEWER#1

General Comments: Overall, all sections are very well-written and designed to describe this novel apparatus. No major comments. I have a few specific comments that are listed below by section and line number.

More Specific Comments:

Abstract

Line 28 – Remove “for performance test”

: We removed “for performance test”

Introduction

Line 49 – Change “penetrated” to “penetrate”

: We corrected “penetrated” to “penetrate”

Results

Line 79 – Reword/rephrase “average of 8168 particles”. I am having difficulty understanding this sentence

: The sentence was reworded to describe the ambiguity.

Discussion – None

Materials and Methods – None

Conclusions – None

Figures and Tables – None

Reviewer 2 Report

PIPAC is very interesting, but there are some points to revise before publish.

1; In “2.2. Distribution analysis with methylene blue solution”, definition of concentrated area is important to evaluate, but this definition is not quantitative, and I think it is not constant. Could you tell me more concentrate definition if authors can?

2; More than half of the paragraph 1 of discussion part is Introduction. Authors should revise this paragraph.

3; I would like to know how much to improve clinical results of PC using this device. Could you tell me about your thought in Discussion section?

Author Response

REVIEWER#2

PIPAC is very interesting, but there are some points to revise before publish.

1; In “2.2. Distribution analysis with methylene blue solution”, definition of concentrated area is important to evaluate, but this definition is not quantitative, and I think it is not constant. Could you tell me more concentrate definition if authors can?

: We have added the definition of “concentrated” and “spread” zones in line 87.

2; More than half of the paragraph 1 of discussion part is Introduction. Authors should revise this paragraph.

: We have revised paragraph 1 of Discussion to avoid redundancy.

3; I would like to know how much to improve clinical results of PC using this device. Could you tell me about your thought in Discussion section?

: We have mentioned the clinical outcomes of PIPAC in the first paragraph in Discussion.

Reviewer 3 Report

The manuscript presents the new prototype for PIPAC and compares it with existent devices. Literature describes numerous cases of PIPAC, but the level of evidence about the efficacy of the method is low so far, since many patients were treated outside clinical trials.

The new device is well described, but evaluation is a little bit less elaborated. Authors aim to overcome problems associated with distribution and penetration of the drugs, offering lower pressure in the system as an advantage.

Although authors compare new device several times with the existing one, the evaluation didn't include any direct comparison in the same setting on the same models. The size of droplets was calculated by equation, not measured in the real experiment (at least measurement was not described). Distribution was insufficiently described: the definition of "concentrated" and "spread" zones was not defined, neither the distance of spraying. The experiment does not explain, how the distribution will be improved outside the spray angle (lateral, proximal walls, in unexposed parts of abdominal cavity). Authors conclude, that DCD is not dependent on nozzle position, however, in all 3 positions was the piece of tissue well in the "concentrated" zone. Would different time intervals after spraying (diffusion time) change the DCD?

Authors conclude, that the new device generates small (not smaller!) droplets and wider spray area than MIP and all of that at lower pressure. However - devices were not directly compared under the same circumstances. The non-varying DCD is perhaps just the reflection of the same time (30 min) left for diffusion. There is no evidence, that the new device increases the homogeneous drug distribution.

Author Response

REVIEWER#3

The manuscript presents the new prototype for PIPAC and compares it with existent devices. Literature describes numerous cases of PIPAC, but the level of evidence about the efficacy of the method is low so far, since many patients were treated outside clinical trials.

The new device is well described, but evaluation is a little bit less elaborated. Authors aim to overcome problems associated with distribution and penetration of the drugs, offering lower pressure in the system as an advantage.

: There has been an increase in clinical demand for PIPAC in recent years, but the availability of the system is mostly limited to European countries. Therefore, we have developed a system expected to have comparable performance to that of MIP. The paper does not aim to overcome problems regarding distribution and penetration of the drugs as mentioned in Discussion (line 137). We would like to emphasize that the prototype has comparable performance to that of the currently available PIPAC system, and that the prototype may offer improved clinical safety as it requires lower pressure generation. We acknowledge that limitations exist regarding drug distribution area and suggest that further studies should address this limitation (line 154). We have revised the last paragraph of Introduction (line 66) to avoid ambiguity.  

Although authors compare new device several times with the existing one, the evaluation didn't include any direct comparison in the same setting on the same models. The size of droplets was calculated by equation, not measured in the real experiment (at least measurement was not described).

: The size of droplets was measured in an experiment using an instrument mentioned in the Method section (line 201). The two equations mentioned in the Results section (line 73) are calculated based on the measurements from the experiment.

Distribution was insufficiently described: the definition of "concentrated" and "spread" zones was not defined, neither the distance of spraying. The experiment does not explain, how the distribution will be improved outside the spray angle (lateral, proximal walls, in unexposed parts of abdominal cavity).

: The distance of spraying is described in the method section (line 208). We have added the definition of “concentrated” and “spread” zones in line 87. In this study, we do not have the intention to demonstrate that the distribution area would be far better than that of the MIP. We have mentioned limitations regarding drug distribution area outside the spray angle in discussion.  

Authors conclude, that DCD is not dependent on nozzle position, however, in all 3 positions was the piece of tissue well in the "concentrated" zone. Would different time intervals after spraying (diffusion time) change the DCD?

: We have included below statement in the conclusion section. Further studies will be required to investigate the comparable effects of the current PIPAC and our prototype.

“We developed a novel prototype that generates small droplets from drug aerosolization and a wider sprayed area compared to the current MIP® at lower pressure. Therefore, this novel prototype is expected to show comparable effects for PIPAC, which should be investigated in further studies”

Authors conclude, that the new device generates small (not smaller!) droplets and wider spray area than MIP and all of that at lower pressure. However - devices were not directly compared under the same circumstances. The non-varying DCD is perhaps just the reflection of the same time (30 min) left for diffusion. There is no evidence, that the new device increases the homogeneous drug distribution.

: We have mentioned the limitation of the prototype regarding homogeneous drug distribution in Discussion (From line 137-158). We suggested mechanical movement of a nozzle in further studies to overcome such limitations.